# Depressive and anxiety symptoms and COVID-19-related factors among men and women in Nigeria

Olakunle Ayokunmi Oginni[1,2☯]*, Ibidunni Olapeju Oloniniyi[1,2☯], Olanrewaju Ibigbami[1,2☯], Victor Ugo[3‡], Ayomipo Amiola[2‡], Adedotun Ogunbajo[4‡], Oladoyin Esan[2‡], Aderopo Adelola[2‡], Oluwatosin Daropale[5‡], Matthew Ebuka[2‡], Boladale Mapayi[1,2‡]

1 Department of Mental Health, Obafemi Awolowo University, Ile-Ife, Nigeria, 2 Mental Health Unit, Obafemi Awolowo University Teaching Hospitals Complex, Ile-Ife, Nigeria, 3 Mentally Aware Nigeria Initiative and Senior Campaign Officer, United for Global Mental Health, London, United Kingdom, 4 Department of Epidemiology, Harvard T.H. Chan School of Public Health, Boston, Massachusetts, United States of America, 5 Health Centre, Joseph Ayo Babalola University, Ikeji, Osun, Nigeria

☯ These authors contributed equally to this work.
‡ These authors also contributed equally to this work.
* kaoginni@yahoo.co.uk

**Data Availability Statement:** Data cannot be shared publicly to preserve anonymity of the study participants. Data are available from the Ethics and Research Committee of the Institute of Public

## Abstract

Despite the greater adverse economic impacts in low and middle-income (LAMI) compared to high-income countries, fewer studies have investigated the associations between COVID-19-related stressor and mental health in LAMI countries. The objectives of this study were to determine the associations between COVID-19-related stressors and anxiety and depressive symptoms while controlling for known risk and protective factors and to investigate any sex differences. An online survey was carried out to assess sociodemographic, psychosocial (previous mental health conditions, sexual orientation, intimate partner violence and perceived social support) and COVID-19-related variables. Hierarchical linear regression was carried out with anxiety and depressive symptoms as separate outcomes. Of the COVID-19-related factors, testing positive for COVID-19 infection, having COVID-19 symptoms, having other medical conditions, self-isolating due to COVID-19 symptoms, worry about infection, perception of the pandemic as a threat to income and isolation during the lockdown were significantly associated with higher anxiety and depressive symptoms. Of these, worry about infection, isolation during lockdown and disruption due to the pandemic retained independent associations with both outcomes. The variance in anxiety and depressive symptoms explained by COVID-19-related factors was larger in women (11.8%) compared to men (6.1% and 0.8% respectively). COVID-19-related stressors are associated with higher anxiety and depressive symptoms, with these effects being larger in men compared to women. Enhancing social support can be an affordable strategy to mitigate this risk but this needs to be investigated using appropriate designs.

Health, Obafemi Awolowo University, Ile-Ife, Nigeria (contact via iphoaurec@gmail) for researchers who meet the criteria for access to confidential data.

**Funding:** The author(s) received no specific funding for this work.

**Competing interests:** The authors have declared that no competing interests exist.

## Introduction

The adverse mental health impacts of coronavirus disease of 2019 (COVID-19) are being increasingly recognized and these include higher rates of depressive and anxiety symptoms [1, 2]. Several mechanisms have been proposed for these associations including neurotoxic effects, although the evidence for this is limited [3, 4]. These adverse mental health problems have also been attributed to the adverse psychosocial effects of the pandemic such as worry about infection—especially among individuals with health conditions associated with increased risk for severe COVID-19 such as diabetes mellitus and hypertension, increased social isolation, disruption of social activities, losses of family members and acquaintances to COVID-19-related mortality and concern about financial instability due to job losses [5].

Alternatively, the increased mental health problems in the context of the COVID-19 pandemic may be partly explained by an exacerbation of pre-existing vulnerabilities rather than independent effects of pandemic-related stress. For example, the public health measures put in place to control the spread of COVID-19—including lockdowns, restricted movement and social contact, isolation and quarantine—may limit access to mental health services for individual in treatment which may in turn exacerbate existing mental health problems. Similarly, the increased tension and frustration resulting from the pandemic and control measures may drive an increase in the rates of intimate violence victimization [6, 7] which is independently associated with depressive and anxiety disorders [8]. Sexual minority individuals, who typically report higher rates of mental health problems including depressive and anxiety disorders [9–11], may also be exposed to greater harassment and discrimination by isolating with family members who are unaccepting of their sexual identity [12]. While these psychosocial vulnerabilities have been shown to be independently associated with higher psychopathology among Nigerians [11, 13–15], their associations with mental health distress in the context of the COVID-19 pandemic has not been previously investigated in Nigeria.

Furthermore, many studies investigating the mental health impacts of the COVID-19 pandemic have largely focused on the risk factors for adverse mental health outcomes with little emphasis on protective factors. This is even more important in low and middle-income countries like Nigeria where mental health coverage and funding are inadequate [16, 17]. Protective factors such as social support [18] may represent a readily available and inexpensive strategy for promoting mental well-being among Nigerians during the COVID-19 pandemic.

The associations between pandemic-related stressors and adverse mental health outcomes have been largely investigated in higher-income countries compared to countries in Africa where the capacities of the governments to cope with the economic impacts of the pandemic is relatively limited. For example, financial assistance to individuals who had to stop working due to the restrictions in physical movement in England will be provided for up to one year [19, 20]. In contrast, the lockdown initiated in Nigeria had to be terminated after one month as the government could not provide sufficient financial support for individuals adversely affected [21]. Considering the pre-existing high levels of poverty and unemployment in Nigeria [21] and reports of adverse impacts of the pandemic on the Nigerian economy [22, 23], it is likely that the associations between COVID-19-related stressors and anxiety and depressive symptoms will be consistent with those previously described in higher-income countries.

Lastly, considering the higher rates of anxiety and depressive disorders observed among women [24] which have been partly attributed to biological and social vulnerabilities including hormonal influences [25] and gender inequalities [26] respectively; it is important to investigate whether women in Nigeria are more vulnerable to the adverse impacts of the COVID-19 pandemic on mental well-being compared to men. This is especially important in sub-Saharan

African countries where gender inequality is very high [27] and may have increased during the pandemic [28].

An understanding of the mechanisms of the adverse impacts of the COVID-19 pandemic on mental health can inform strategies to minimize these adverse outcomes. Protective factors can be promoted while identified risk factors can be used to selectively target those at highest risk for preventive or therapeutic interventions. The objectives of this study were, therefore, to identify sociodemographic, psychosocial and COVID-19-related factors associated with anxiety and depressive symptoms among Nigerians in the wake of the COVID-19 pandemic; and to identify sex differences in these relationships. We hypothesized that COVID-related stressors will be independently associated with anxiety and depressive symptoms and that these associations will be stronger in women.

## Materials and methods

This online quantitative survey recruited Nigerians who were at least 18 years old, resident in Nigeria for at least six months prior to the lockdown, fluent in English, able to use the internet and reported no severe cognitive or physical impairments. These criteria were included in the survey as single questions and participants who did not meet them were excluded from the survey. The survey tool consisted of a participant information page explaining the aims and objectives of the study, an informed consent page, and the survey. The survey was posted on social media including Facebook, Twitter and WhatsApp groups; and male and female participants had equal chances of participating in the survey.

The study sample comprised individuals from all parts of Nigeria (Yorubas -70.0%, Igbos– 17.8%, Hausas– 4.5%, and other tribes– 7.6%). Of the 1013 individuals who participated in the study, 43 were excluded due to high levels of missing data while 4 individuals identified as being gender non-binary and were excluded from analyses by sex as their small number precluded further sub-analyses. This gave a total 966 participants which was larger and comparable to other online surveys which were carried out in Nigeria during the COVID-19 pandemic [29, 30]. Ethical approval was obtained from the Ethics and Research Committee of the Institute of Public Health, Obafemi Awolowo University, Ile-Ife, Nigeria. Online informed consent was obtained from all study participants.

### Measures

The sociodemographic section elicited the sex and age of the participants in years, the highest level of education with options including *'No qualifications'*, *'Primary school'*, *'Vocational training and equivalents'*, *'Secondary school'* and *'University'*. Marital status was ascertained as *'Single'*, *'Married'* and *'Divorced, Separated or widowed'*. Employment status categorized as *'Unemployed/Retired'*, *'Students'* and *'Employed'*. For those who were employed, income was categorized into ≤₦40,000 (≤USD105), ₦40,001–80,000 (USD105-210), ₦80,001–120,000 (USD210-315), ₦120,001–160,000 (USD315-420) and >₦160,000 (USD420) to limit the effect of skewness. These questions have been previously used to assess sociodemographic information among Nigerians [11, 31] and were therefore considered appropriate.

Psychosocial stressors known to be associated with anxiety and depressive symptoms were assessed as follows: A single question was asked about whether the participants had experienced mental health problems prior to the COVID-19 pandemic. Responses were either *'Yes'* or *'No'*. Participants were asked to indicate their sexual orientation and the responses were categorized as *'Heterosexual'* and *'Non-heterosexual'* (comprising *'Mostly Heterosexual'*, *'Bisexual'*, *'Mostly gay'* and *'Completely gay'*). Intimate Partner violence was assessed using the HARK questionnaire which comprises four questions each screening for past-year physical, sexual

and emotional abuse [32]. Each item was scored 0 (No) or 1 (Yes) and the sum of the scores used in subsequent analyses, Cronbach's alpha in the present study was 0.73. Perceived social support was assessed using the 12-item Multidimensional Scale of Perceived Social Support [33]. Each item was rated on a 7-point Likert scale ranging from 1 (Very strongly disagree) to 7 (Very strongly agree) and total scores derived as a sum of individual responses were used in analyses. Higher scores indicated higher perceived support and Cronbach's alpha in the present study was 0.97.

Stressors related to the COVID-19 pandemic were assessed using questions adapted from a previous survey [34]. These included whether participants i. had previously tested positive for COVID-19, ii. had had any COVID-19 symptoms, iii. had any medical conditions including hypertension and diabetes mellitus that could increase their risk for COVID-19 infection, iv. had had to self-isolate due to COVID-19 symptoms, v. had close friends who had tested positive for COVID-19, and vi. knew any persons who had died from COVID-19 infection. These were assessed using single questions with *'Yes'* or *'No'* responses. Participants' worry about themselves and their family members becoming infected with COVID-19 was elicited with two separate questions with responses scored on a 5-point Likert scale ranging from *'Not worried at all'* (scored 1) to *'Extremely worried'* (scored 5). Two single questions were asked about whether the participants perceived the COVID-19 pandemic as a threat to income and whether they had felt isolated during the lockdown. These were scored on a Likert scale ranging from 0 (Strongly disagree) to 4 (Strongly agree). Disruption due to the COVID-19 pandemic on domains of life including family, marriage/intimate relationships, parenting, friendships/social life, work, education, recreation, spirituality, community life and physical self-care was assessed by asking participants to indicate how much the pandemic had impacted on each of these domains. Responses for each domain ranged from 1 (Not at all) to 10 (Serious disruption). A total score was computed by summing the individual responses and used in subsequent analyses with higher scores indicating greater disruption; the Cronbach's alpha in the present study was 0.91.

The outcome variables comprised anxiety and depressive symptoms which were assessed via seven items each from the 14-item Hospital and Anxiety Scale (HADS) [35]. Each item was rated on a 4-point Likert scale ranging from 0 (No, not at all) to 3 (Yes, definitely). Total scores derived for the anxiety and depressive sub-scales by summing the responses were used in analyses and higher scores indicated more severe symptoms; Cronbach's alphas were 0.81 and 0.64 respectively.

## Analyses

Analyses were carried out using STATA software (vs 14). Data were summarized using proportions, means (and standard deviations) and medians (and interquartile ranges) as appropriate; while sex differences tested using Chi-square test, independent samples t-test and Wilcoxon rank sum test respectively. Univariate associations were investigated using linear regression with anxiety and depressive symptom scores as outcome variables and sociodemographic, COVID-19-related and psychosocial factors as predictor variables. The unstandardized coefficients and their 95% confidence intervals, and the standardized coefficients (beta) were reported. Hierarchical multivariate linear regression models were subsequently specified for anxiety and depressive symptoms as outcome variables respectively while predictor variables included socio-demographic, psychosocial and COVID-19-related variables. These enabled the identification of variables independently associated with anxiety and depressive symptoms. Only variables with univariate associations at $p<0.10$ with each outcome were included in the multivariate models. Socio-demographic variables were included in the first

step, psychosocial stressors included in the second step and the COVID-19-related variables included in the third step. We reported the change in variance ($\Delta R^2$) of the outcome variable with the addition of each block of variables at each step and the regression coefficients for the final model containing the socio-demographic, psychosocial and COVID-19-related variables. Statistical significance of the change in variance explained was determined using likelihood ratio tests at $p<0.05$.

All steps of the analyses were carried out in the whole sample and in males and females to determine any sex differences. For the univariate and multivariate regression analyses, sex differences in the regression coefficients were determined by inspecting the 95% confidence intervals of the coefficients in male and female participants and differences were deemed to be statistically significant when the confidence intervals did not overlap. Similarly, sex differences in $\Delta R^2$ in multivariate regression models were determined by comparing the magnitudes of $\Delta R^2$ in males and female participants.

## Results

### Descriptive statistics and sex differences

The participants' median age was 29.0 (IQR: 12.0) years with female participants being significantly younger (median: 27.0, IQR: 11.0 years) compared to males (median: 31.0, IQR: 13.0 years; Table 1). Majority (78.7%) had a university qualification with male participants having significantly lower educational qualifications. More respondents were single (61%) and employed (58.1%) with males being more likely to be married and employed ($\chi^2$ = 19.64 and 9.70 respectively, $p < 0.01$ and 0.001 respectively). Of those who were employed, about a third (31.4%) earned a monthly income ranging between ₦41,000 and ₦80,000 (209.2 USD) and these proportions were comparable in both sexes.

Five percent of the participants reported having experienced mental health problems prior to the COVID-19 pandemic and 22% indicated being non-heterosexual, there were no sex differences in these proportions. Thirty percent of the participants had experienced at least one form of intimate partner violence and this was significantly higher among women compared to men ($\chi^2$ = 15.93; $p < 0.01$). The mean anxiety symptom scores were comparable in men and women while depressive symptom and perceived support scores were significantly higher in women compared to men (t = 2.00, $p > 0.05$; Wilcoxon rank sum coefficient = 4.90, $p < 0.001$ respectively).

About 5% of the participants reported testing positive for COVID-19 infection while 12.6% had experienced COVID-19 symptoms, 4% had COVID-19 risk conditions, 13.2% had had to self-isolate because of COVID-19 symptoms, 15.6% reported having friends who had tested positive for COVID-19 infection, 17.7% knew someone who had died from COVID-19 infection and these rates were comparable in men and women. However, compared to men, women were significantly more worried about themselves and family members getting infected with the COVID-19 ($\chi^2$ = 60.42 and 48.00 respectively; $p < 0.001$) and were more likely to see the pandemic as a threat to income ($\chi^2$ = 14.55; $p < 0.01$). In contrast, men reported higher disruption from the pandemic compared to women (Wilcoxon rank sum coefficient = 2.34, $p < 0.05$).

### Univariate associations and sex differences

**Anxiety symptoms.** Of the socio-demographic variables, being employed was associated with significantly less anxiety symptoms compared to those who were unemployed/retired (β = -1.03, 95% CI: -3.17, -0.16) and earning a higher income was associated with lower anxiety symptoms in the whole sample (Table 2). Among male participants, higher age was

**Table 1. Sociodemographic, COVID-19-related and psychosocial factors with sex differences.**

| VARIABLES | Total sample | % | Male | % | Female | % | Statistic |
|---|---|---|---|---|---|---|---|
| | n = 966 | % | n = 487 | % | n = 479 | % | Chi-square/t-test/Wilcoxon signed rank test |
| **SOCIO-DEMOGRAPHIC VARIABLES** | | | | | | | |
| **Age (Median and IQR)** | 29.0 | 12.0 | 31.0 | 13.0 | 27.0 | 11.0 | -7.32***[b] |
| **Level of education** | | | | | | | |
| No qualifications | 21 | 2.2 | 14 | 2.87 | 6 | 1.3 | 20.01*** |
| Primary School | 21 | 2.2 | 13 | 2.67 | 8 | 1.7 | |
| Vocational and equivalents | 55 | 5.7 | 40 | 8.21 | 14 | 2.9 | |
| Secondary school | 110 | 11.3 | 46 | 9.45 | 64 | 13.4 | |
| University | 763 | 78.7 | 374 | 76.80 | 387 | 80.8 | |
| **Marital status** | | | | | | | |
| Single | 592 | 61.0 | 268 | 55.03 | 321 | 67.0 | 19.64*** |
| Married | 338 | 34.9 | 203 | 41.68 | 135 | 28.2 | |
| Divorced, Separated or Widowed | 40 | 4.1 | 16 | 3.29 | 23 | 4.8 | |
| **Employment status** | | | | | | | |
| Unemployed/Retired | 126 | 13.0 | 61 | 12.53 | 65 | 13.6 | 9.70** |
| Student | 280 | 28.9 | 120 | 24.64 | 158 | 33.0 | |
| Employed | 564 | 58.1 | 306 | 62.83 | 256 | 53.4 | |
| **Income (₦,000) (n = 549)** | | | | | | | |
| ≤40 | 110 | 20.04 | 56 | 18.98 | 54 | 21.43 | 1.30 |
| 41–80 | 178 | 32.42 | 94 | 31.86 | 83 | 32.94 | |
| 81–120 | 75 | 13.66 | 44 | 14.92 | 31 | 12.30 | |
| 121–160 | 42 | 7.65 | 24 | 8.14 | 18 | 7.14 | |
| >160 | 144 | 26.23 | 77 | 26.10 | 66 | 26.19 | |
| **PSYCHOSOCIAL FACTORS** | | | | | | | |
| **History of mental health problems** | | | | | | | |
| No | 902 | 95.0 | 449 | 94.3 | 449 | 95.5 | 0.71 |
| Yes | 48 | 5.1 | 27 | 5.7 | 21 | 4.5 | |
| **Sexual orientation** | | | | | | | |
| Heterosexual | 756 | 77.9 | 380 | 78.0 | 376 | 78.5 | 0.03 |
| Non-heterosexual | 214 | 22.1 | 107 | 22.0 | 103 | 21.5 | |
| **Intimate partner violence** | | | | | | | |
| 0 | 674 | 69.5 | 362 | 74.3 | 311 | 64.9 | 15.93** |
| 1 | 106 | 10.9 | 49 | 10.1 | 57 | 11.9 | |
| 2 | 117 | 12.1 | 52 | 10.7 | 63 | 13.2 | |
| 3 | 39 | 4.0 | 16 | 3.3 | 22 | 4.6 | |
| 4 | 34 | 3.5 | 8 | 1.6 | 26 | 5.4 | |
| **Perceived social support (Median and IQR)** | 39.0 | 32.0 | 36.0 | 31.0 | 43.0 | 31.0 | 4.90***[b] |
| **Anxiety symptoms (Mean and SD)** | 15.8 | 4.50 | 15.6 | 4.71 | 16.0 | 4.28 | 1.31[a] |
| **Depressive symptoms (Mean and SD)** | 14.8 | 3.55 | 14.6 | 3.53 | 15.0 | 3.56 | 2.00*[a] |
| **COVID-19 RELATED VARIABLES** | | | | | | | |
| **Tested positive for COVID-19** | | | | | | | |
| No | 923 | 95.2 | 458 | 94.1 | 461 | 96.2 | 2.52 |
| Yes | 47 | 4.9 | 29 | 6.0 | 18 | 3.8 | |
| **COVID-19 symptoms** | | | | | | | |
| No | 848 | 87.4 | 423 | 86.9 | 423 | 88.3 | 0.47 |
| Yes | 122 | 12.6 | 64 | 13.1 | 56 | 11.7 | |

*(Continued)*

**Table 1.** (Continued)

| VARIABLES | Total sample | % | Male | % | Female | % | Statistic |
|---|---|---|---|---|---|---|---|
| | n = 966 | % | n = 487 | % | n = 479 | % | Chi-square/t-test/Wilcoxon signed rank test |
| **Other medical conditions** | | | | | | | |
| No | 928 | 95.7 | 460 | 94.5 | 464 | 96.9 | 3.38† |
| Yes | 42 | 4.3 | 27 | 5.5 | 15 | 3.1 | |
| **Self-Isolation for COVID -19 Symptoms** | | | | | | | |
| No | 842 | 86.8 | 425 | 87.3 | 415 | 86.6 | 0.08 |
| Yes | 128 | 13.2 | 62 | 12.7 | 64 | 13.4 | |
| **Having friend who had COVID-19** | | | | | | | |
| No | 819 | 84.4 | 407 | 84.2 | 407 | 85.0 | 0.11 |
| Yes | 151 | 15.6 | 77 | 15.0 | 72 | 15.0 | |
| **Knowing someone who died from COVID-19 infection** | | | | | | | |
| No | 798 | 82.3 | 406 | 83.4 | 389 | 81.2 | 0.77 |
| Yes | 172 | 17.7 | 81 | 16.6 | 90 | 18.8 | |
| **Worry about COVID-19 infection (n = 952)** | | | | | | | |
| Not worried at all | 228 | 24.0 | 165 | 34.5 | 63 | 13.4 | 60.42*** |
| Slightly worried | 234 | 24.6 | 106 | 22.1 | 124 | 26.4 | |
| Somewhat worried | 173 | 18.2 | 74 | 15.5 | 99 | 21.1 | |
| Moderately worried | 193 | 20.3 | 75 | 15.7 | 118 | 25.2 | |
| Extremely worried | 124 | 13.0 | 59 | 12.3 | 65 | 13.9 | |
| **Worry COVID-19 infection in family members (n = 965)** | | | | | | | |
| Not worried at all | 172 | 17.8 | 121 | 24.9 | 51 | 10.8 | 48.00*** |
| Slightly worried | 223 | 23.1 | 120 | 24.6 | 101 | 21.3 | |
| Somewhat worried | 162 | 16.8 | 74 | 15.2 | 87 | 18.4 | |
| Moderately worried | 181 | 18.8 | 63 | 12.9 | 118 | 25.0 | |
| Extremely worried | 227 | 23.5 | 109 | 22.4 | 117 | 24.7 | |
| **COVID-19 pandemic perceived as threat to income** | | | | | | | |
| Strongly disagree | 105 | 10.8 | 68 | 14.0 | 37 | 7.7 | 14.55** |
| Slightly disagree | 174 | 17.9 | 82 | 16.8 | 90 | 18.8 | |
| Neutral | 183 | 18.9 | 102 | 20.9 | 81 | 16.9 | |
| Slightly agree | 213 | 22.0 | 97 | 19.9 | 116 | 24.2 | |
| Strongly agree | 295 | 30.4 | 138 | 28.3 | 155 | 32.4 | |
| **Feeling isolated during lockdown** | | | | | | | |
| Strongly disagree | 127 | 13.1 | 72 | 14.8 | 55 | 11.5 | 5.74 |
| Slightly disagree | 208 | 21.4 | 99 | 20.3 | 106 | 22.1 | |
| Neutral | 233 | 24.0 | 123 | 25.3 | 109 | 22.8 | |
| Slightly agree | 249 | 25.7 | 113 | 23.2 | 136 | 28.4 | |
| Strongly agree | 153 | 15.8 | 80 | 16.4 | 73 | 15.2 | |
| **Disruption from COVID-19 pandemic (Median and IQR)** | 34.0 | 27.0 | 36.0 | 27.0 | 32.0 | 26.0 | 2.37*b |

[a]T-test,

[b]Wilcoxon rank sum test;

† $p < 0.1$;

* $p < 0.05$;

** $p < 0.01$;

*** $p < 0.001$

**Table 2. Univariate associations between anxiety symptoms and predictors with sex differences.**

| | Total | | | | Male | | | | Female | | |
|---|---|---|---|---|---|---|---|---|---|---|---|
| Predictor variables | Coef | 95% CI | Beta | | Coef | 95% CI | Beta | | Coef | 95% CI | Beta |
| **SOCIO-DEMOGRAPHIC FACTORS** | | | | | | | | | | | |
| Age | 0.00 | -0.28, 0.29 | 0.00 | | 0.04* | 0.01, 0.08 | 0.10 | | -0.06* | -0.10, -0.01 | -0.11 |
| **Education (Ref = No qualifications)** | | | | | | | | | | | |
| Primary School | 0.71 | -1.97, 3.40 | 0.02 | | 0.65 | -2.80, 4.10 | 0.02 | | -5.13 | -4.54, 4.54 | 0.00 |
| Vocational and equivalents | 1.08 | -1.15, 3.31 | 0.06 | | 0.30 | -2.48, 3.08 | 0.02 | | 1.36 | -2.75, 5.46 | 0.05 |
| Secondary school | -1.83† | -3.90, 0.25 | -0.13 | | -3.24* | -5.97, -0.51 | -0.20 | | -0.66 | -4.25, 2.94 | -0.05 |
| University | -1.89† | -3.81, 0.04 | -0.17 | | -3.39† | -5.83, -0.96 | -0.30 | | -0.55 | -4.01, 2.91 | -0.05 |
| **Marital status (Ref = Single)** | | | | | | | | | | | |
| Married | -0.24 | -0.85, 0.36 | -0.03 | | 0.74† | -0.11, 1.60 | 0.08 | | -1.33* | -2.19, -0.48 | -0.14 |
| Divorced, Separated or Widowed | 0.96 | -0.48, 2.41 | 0.04 | | 1.80 | -0.57, 4.18 | 0.07 | | 0.32 | -0.57, 4.18 | 0.02 |
| **Employment status (Ref = Unemployed/Retired)** | | | | | | | | | | | |
| Student | -0.01 | -0.95, 0.93 | 0.00 | | 0.35 | -1.11, 1.81 | 0.03 | | -0.40 | -1.61, 0.82 | -0.04 |
| Employed | -1.02* | -1.89, -0.16 | -0.11 | | -0.02 | -1.32, 1.28 | 0.00 | | -2.01** | -3.15, -0.86 | -0.23 |
| **Income (₦,000; Ref: <40)** | | | | | | | | | | | |
| 41–80 | 0.22 | -0.82, 1.26 | 0.02 | | 0.09 | -1.37, 1.54 | 0.01 | | 0.28 | -1.17, 1.73 | 0.03 |
| 81–120 | -1.04 | -2.32, 0.25 | -0.08 | | -2.20 | -3.94, -0.46 | -0.17 | | 0.26 | -1.61, 2.13 | 0.02 |
| 121–160 | -1.62* | -3.17, -1.03 | -0.10 | | -2.35* | -4.45, -0.24 | -0.14 | | -0.93 | -3.19, 1.33 | -0.06 |
| >160 | -2.11*** | -3.20, -1.03 | -0.21 | | -3.98*** | -5.50, -2.47 | -0.37 | | -0.11 | -1.63, 1.42 | -0.01 |
| **PSYCHOSOCIAL FACTORS** | | | | | | | | | | | |
| **History of mental health problems (Ref = No)** | 2.09** | 0.79, 3.39 | 0.10 | | 2.14* | 0.32, 3.96 | 0.11 | | 2.07* | 0.20, 3.93 | 0.10 |
| **Sexual orientation (Ref = Heterosexual)** | 0.72* | 0.38, 1.40 | 0.07 | | 0.28 | -0.73, 1.30 | 0.02 | | 1.22* | 0.29, 2.15 | 0.12 |
| **Intimate partner violence** | 0.89*** | 0.63, 1.15 | 0.21 | | 1.03*** | 0.59, 1.48 | 0.20 | | 0.77*** | 0.45, 1.09 | 0.21 |
| **Perceived social support** | -0.10*** | -0.11, -0.08 | -0.37 | | -0.11*** | -0.14, -0.09 | -0.41 | | -0.09*** | -0.11, -0.07 | -0.35 |
| **COVID-RELATED FACTORS** | | | | | | | | | | | |
| **Tested positive for COVID-19 (Ref = No)** | 2.61*** | 1.30, 3.92 | 0.12 | | 3.29*** | 1.54, 5.04 | 0.17 | | 1.66 | -0.35, 3.68 | 0.07 |
| **COVID-19 symptoms (Ref = No)** | 1.90*** | 1.05, 2.74 | 0.14 | | 2.17* | 0.94, 3.40 | 0.16 | | 1.57* | 0.38, 2.75 | 0.12 |
| **Other medical conditions (Ref = No)** | 2.01** | 0.62, 3.40 | 0.09 | | 2.86** | 1.05, 4.68 | 0.14 | | 0.68 | -1.53, 2.88 | 0.03 |
| **Self-isolation for COVID-19 (Ref = No)** | 1.71*** | 0.87, 2.54 | 0.13 | | 1.90** | 0.65, 2.15 | 0.13 | | 1.50** | 0.38, 2.62 | 0.12 |
| **Having friend who had COVID-19 (Ref = No)** | -0.04 | -0.82, 0.74 | 0.00 | | 0.27 | -0.88, 1.42 | 0.02 | | -0.45 | -1.53, 0.62 | -0.04 |
| **Knowing someone who died from COVID-19 (Ref = No)** | 0.37 | -0.37, 1.11 | 0.03 | | 0.44 | -0.68, 1.57 | 0.04 | | 0.23 | -0.75, 1.22 | 0.02 |
| **Worry about COVID-19 infection** | 0.42*** | 0.21, 0.62 | 0.13 | | 0.09 | -0.21, 0.39 | 0.03 | | 0.80*** | 0.50, 1.10 | 0.24 |
| **Worry COVID-19 infection in family members** | 0.17† | -0.03, 0.37 | 0.05 | | -0.12 | -0.40, 0.16 | -0.04 | | 0.53*** | 0.24, 0.81 | 0.16 |
| **COVID-19 pandemic perceived as threat to income** | -0.43*** | -0.64, -0.23 | -0.13 | | -0.76*** | -1.06, -0.47 | -0.23 | | -0.07 | -0.36, 0.22 | -0.03 |
| **Feeling isolated during lockdown** | -0.14 | -0.36, 0.09 | -0.04 | | -0.88*** | -1.19, -0.56 | -0.24 | | 0.68*** | 0.38, 0.98 | 0.20 |
| **Disruption from COVID-19 pandemic** | 0.00 | -0.02, 0.02 | 0.00 | | -0.03† | -0.06, 0.00 | -0.09 | | 0.02† | -0.00, 0.05 | 0.09 |

†$p < 0.1$;

*$p < 0.05$;

**$p < 0.01$;

***$p < 0.001$;

Beta—standardized regression coefficient.

significantly associated with higher anxiety symptoms (β = 0.04, 95% CI: 0.01, 0.08) while higher educational qualifications were associated with lower anxiety symptoms. Among females participants, lower age (β = -0.06, 95% CI: -0.10, -0.01), being married (β = -1.33, 95% CI: -2.19, -0.48) and being employed (β = -2.01, 95% CI: -3.15, -0.86) were significantly associated with lower anxiety scores.

Previous diagnoses of mental health problems (β = 2.09, 95% CI: 0.79, 3.39), intimate partner violence (β = 0.89, 95% CI: 0.63, 1.15) and non-heterosexuality (β = 0.72, 95% CI: 0.38, 1.40) were significantly associated with higher anxiety symptoms while higher perceived social support was associated with lower anxiety symptoms in the whole sample (β = -0.10, 95% CI: -0.11, -0.08) and these effects were comparable in male and female participants.

COVID-19 related factors associated with higher anxiety symptoms included testing positive for COVID-19 (β = 2.61, 95% CI: 1.30, 3.92), having COVID-19 symptoms (β = 1.90, 95% CI: 1.05, 2.74), having other medical conditions (β = 2.01, 95% CI: 0.62, 3.40) and self-isolating due to COVID-19 symptoms (βs = 1.71; 95% CIs: 0.87, 2.54) with these associations being larger in men compared to women. The associations between worrying about COVID-19 infection—personally and for family members—and anxiety symptoms were larger in female (βs = 0.80 and 0.53; 95% CIs: 0.50, 1.10 and 0.24, 0.81 respectively) compared to male participants (βs = 0.09 and -0.12; 95% CIs: -0.21, 0.39 and -0.40, 0.16 respectively). While these differences were not statistically significant as indicated by overlapping confidence intervals, feeling isolated during the lockdown was significantly associated with lower anxiety symptoms in men (β = -0.88, 95% CI: -1.19, -0.56) and higher anxiety symptoms in women (β = 0.68, 95% CI: 0.38, 0.98).

**Depressive symptoms.** Higher education and income were significantly associated with fewer depressive symptoms, as were being married (β = -0.57; 95% CIs: -1.05, -0.10) and employed (β = -1.05; 95% CIs: -1.74, -0.37; Table 3); and these effects were comparable in male and female participants.

Previous mental health problems, non-heterosexual sexual orientation and intimate partner violence were associated with higher depressive symptoms (βs = 2.27, 0.60 and 0.64, 95% CIs: 1.25, 3.28; 0.06, 1.13 and 0.44, 0.85 respectively) while perceived social support was associated with less depressive symptoms (β = -0.07, 95% CI: -0.09, -0.06). Although these effects were larger in women compared to men, these differences were not statistically significant.

Self-isolation due to COVID-19 symptoms, worry about getting infected and feeling isolated were associated with higher depressive symptoms (βs = 0.70, 0.24 and 0.29; 95% CIs: 0.04, 1.36, 0.07, 0.40 and 0.11, 0.46 respectively). However, though all effects appeared larger in women compared to men, these differences were not statistically different.

## Multivariate associations and sex differences

**Anxiety symptoms.** Sociodemographic variables explained 6.1% of the variance in anxiety symptoms while female sex and lower levels of education were independently associated with higher anxiety symptoms (Table 4). None of the other sociodemographic variables were independently associated with anxiety symptoms in men and women, however, the direction of effects remained consistent. The variance explained by socio-demographic variables was larger in men compared to women (9.2% and 5.9% respectively).

Psychosocial factors explained the highest proportion of the variance in anxiety symptoms (14.7% in the whole sample, and 16.0% and 13.6% among men and women respectively). Of these variables, perceived social support was independently associated with lower anxiety while intimate partner violence was further associated with higher anxiety symptoms and this pattern was consistent in men and women.

COVID-19-related factors explained 5.0% of the variance in anxiety symptoms, with worry about infection emerging as the single independent predictor in the whole sample. The variance explained by COVID-19 related factors was larger in female (11.6%) compared to male participants (6.1%). Among men and women, disruption due to the pandemic was

**Table 3. Univariate associations between depressive symptoms and predictors with sex differences.**

| | Total | | | Male | | | Female | | |
|---|---|---|---|---|---|---|---|---|---|
| Predictor variables | Coef | 95% CI | Beta | Coef | 95% CI | Beta | Coef | 95% CI | Beta |
| **SOCIO-DEMOGRAPHIC FACTORS** | | | | | | | | | |
| **Age** | -0.03* | -0.05, 0.00 | -0.08 | -0.01 | -0.04, 0.02 | -0.04 | -0.04* | -0.08, 0.00 | -0.10 |
| **Education (Ref = No qualifications)** | | | | | | | | | |
| Primary School | -0.29 | -2.42, 1.85 | -0.01 | -1.30 | -3.93, 1.33 | -0.06 | 0.63 | -3.14, 4.39 | 0.02 |
| Vocational and equivalents | -0.62 | -2.39, 1.15 | -0.04 | -1.44 | -3.56, 0.68 | -0.11 | 0.21 | -3.19, 3.62 | 0.01 |
| Secondary School | -1.66* | -3.31, -0.02 | -0.15 | -1.88† | -3.97, 0.20 | -0.16 | -1.77 | -4.74, 1.21 | -0.17 |
| University | -2.02* | -3.55, -0.49 | -0.23 | -2.92* | -4.78, -1.06 | -0.35 | -1.55 | -4.43, 1.32 | -0.17 |
| **Marital status (Ref = Single)** | | | | | | | | | |
| Married | -0.57* | -1.05, -0.10 | -0.08 | -0.26 | -0.91, 0.38 | -0.04 | -0.84* | -1.55, -0.12 | -0.11 |
| Divorced, Separated or Widowed | 0.58 | -0.56, 1.71 | 0.03 | 0.66 | -1.13, 2.44 | 0.03 | 0.37 | -1.14, 1.87 | 0.02 |
| **Employment status (Ref = Unemployed/Retired)** | | | | | | | | | |
| Student | -.23 | -0.97, 0.51 | -0.03 | 0.11 | -0.98, 1.19 | 0.01 | -0.57 | -1.59, 0.45 | -0.07 |
| Employed | -1.05** | -1.74, -0.37 | -0.15 | -0.57 | -1.54, 0.40 | -0.08 | -1.50* | -2.47, -0.54 | -0.21 |
| **Income (₦,000; Ref: <40)** | | | | | | | | | |
| 41–80 | 0.15 | -0.69, 0.99 | 0.02 | -0.12 | -1.27, 1.02 | -0.02 | 0.39 | -0.86, 1.64 | 0.05 |
| 81–120 | -0.53 | -1.56, 0.51 | -0.05 | -0.89 | -2.25, 0.48 | -0.09 | -0.13 | -1.74, 1.48 | -0.01 |
| 121–160 | -1.00 | -2.26, 0.25 | -0.07 | -1.29 | -2.94, 0.36 | -0.10 | -0.70 | -2.65, 1.24 | -0.05 |
| >160 | -1.43** | -2.30, -0.55 | -0.18 | -2.28*** | -3.47, -1.09 | -0.28 | -0.51 | -1.82, 0.80 | -0.06 |
| **PSYCHOSOCIAL FACTORS** | | | | | | | | | |
| **History of mental health problems (Ref = No)** | 2.27*** | 1.25, 3.28 | 0.14 | 1.69* | 0.35, 3.03 | 0.11 | 3.07* | 1.54, 4.61 | 0.18 |
| **Sexual orientation (Ref = Heterosexual)** | 0.60* | 0.06, 1.13 | 0.07 | 0.12 | -0.64, 0.88 | 0.01 | 1.09** | 0.32, 1.87 | 0.13 |
| **Intimate Partner Violence** | 0.64*** | 0.44, 0.85 | 0.19 | 0.65*** | 0.31, 0.98 | 0.17 | 0.60*** | 0.34, 0.87 | 0.20 |
| **Perceived Social Support** | -0.07*** | -0.09, -0.06 | -0.36 | -0.08*** | -0.10, -0.06 | -0.39 | -0.08*** | -0.09, -0.06 | -0.36 |
| **COVID-RELATED FACTORS** | | | | | | | | | |
| **Testing positive for Covid-19 (Ref = No)** | 0.48 | -0.56, 1.52 | 0.03 | 0.68 | -0.65, 2.01 | 0.05 | 0.32 | -1.36, 2.00 | 0.02 |
| **COVID-19 symptoms (Ref = No)** | 0.38 | -0.29, 1.06 | 0.04 | 0.51 | -0.42, 1.44 | 0.05 | 0.21 | -0.78, 1.21 | 0.02 |
| **Other medical conditions (Ref = No)** | 0.98† | -.11, 2.08 | 0.06 | 1.36† | -0.01, 2.73 | 0.03 | 0.52 | -1.31, 2.36 | 0.09 |
| **Self-isolation for COVID -19 symptoms (Ref = No)** | 0.70* | 0.04, 1.36 | 0.07 | 0.51 | -0.43, 1.45 | 0.05 | 0.82† | -0.12, 1.76 | 0.08 |
| **Having friend who had COVID-19 (Ref = No)** | -0.56† | -1.18, 0.06 | -0.06 | -0.91* | -1.77, -0.05 | -0.09 | -0.24 | -1.14, 0.65 | -0.02 |
| **Knowing someone who died from COVID-19 (Ref = No)** | -0.34 | -0.93, 0.25 | -0.04 | -0.36 | -1.20, 0.49 | -0.04 | -0.39 | -1.21, 0.43 | -0.04 |
| **Worry about COVID-19 infection** | 0.24** | 0.07, 0.40 | 0.09 | 0.08 | -0.15, 0.30 | 0.03 | 0.39* | 0.13, 0.64 | 0.14 |
| **Worry COVID-19 infection in family members** | 0.01 | -0.15, 0.16 | 0.00 | -0.12 | -0.33, 0.09 | -0.05 | 0.12 | -0.18, 0.36 | 0.05 |
| **COVID-19 pandemic perceived as threat to income** | -0.17* | -0.33, 0.00 | -0.06 | -0.27* | -0.49, -0.04 | -0.11 | -0.07 | -0.31, 0.17 | -0.03 |
| **Feeling isolated during lockdown** | 0.29** | 0.11, 0.46 | 0.10 | -0.14 | -0.38, 0.11 | -0.05 | 0.75*** | 0.51, 1.00 | 0.26 |
| **Disruption from COVID 19 pandemic** | 0.01 | -0.01, 0.02 | 0.03 | -0.01 | -0.03, 0.01 | -0.05 | 0.02* | 0.00, 0.05 | 0.10 |

†$p < 0.1$;

*$p < 0.05$;

**$p < 0.01$;

***$p < 0.001$;

Beta—standardized regression coefficient.

significantly associated with higher anxiety symptoms; however, while isolation was associated with fewer anxiety symptoms in men, it was associated with more anxiety symptoms on women. The presence of COVID-19 symptoms was further associated with more anxiety symptoms in men.

**Table 4. Multivariate associations between anxiety symptoms and predictors with sex differences.**

| Predictor variables | Total | | | Male | | | Female | | |
|---|---|---|---|---|---|---|---|---|---|
| | Coef | 95% CI | Beta | Coef | 95% CI | Beta | Coef | 95% CI | Beta |
| **SOCIO-DEMOGRAPHIC FACTORS ($\Delta R^2$ in %)** | (6.1***) | | | (9.2***) | | | (5.9**) | | |
| **Sex (Ref = Female)** | | | | | | | | | |
| Male | -0.60* | -1.15, -0.07 | -0.07 | | | | | | |
| **Age** | 0.00 | -0.04, 0.04 | -0.01 | -0.01 | -0.07, 0.04 | -0.03 | -0.04 | -0.09, 0.02 | -0.08 |
| **Education (Ref = No qualifications)** | | | | | | | | | |
| Primary School | 0.14 | -2.38, 2.66 | 0.00 | 0.60 | -2.46, 3.66 | 0.02 | -0.27 | -4.84, 4.30 | -0.01 |
| Vocational and equivalents | 0.80 | -1.33, 2.93 | 0.04 | 1.13 | -1.40, 3.65 | 0.07 | -0.51 | -4.64, 3.61 | -0.02 |
| Secondary School | -2.15* | -4.18, -0.13 | -0.15 | -1.79 | -4.27, 0.69 | -0.11 | -1.47 | -5.27, 2.33 | -0.12 |
| University | -1.62† | -3.49, 0.26 | -0.15 | -1.80 | -4.02, 0.43 | -0.16 | -0.46 | -4.11, 3.18 | -0.04 |
| **Marital status (Ref = Single)** | | | | | | | | | |
| Married | -0.02 | -0.80, 0.77 | 0.00 | 0.54 | -0.59, 1.67 | 0.06 | 0.07 | -0.98, 1.13 | 0.01 |
| Divorced, Separated or Widowed | 0.58 | -0.92, 2.08 | 0.02 | 1.43 | -1.01, 3.86 | 0.05 | 0.84 | -0.95, 2.64 | 0.04 |
| **Employment status (Ref = Unemployed/Retired)** | | | | | | | | | |
| Student | 0.69 | -0.24, 1.63 | 0.07 | 0.92 | -0.44, 2.28 | 0.08 | 0.23 | -1.00, 1.45 | 0.02 |
| Employed | -0.64 | -1.46, 0.18 | -0.07 | 0.03 | -1.16, 1.22 | 0.00 | -1.17* | -2.24, -0.09 | -0.14 |
| **PSYCHOSOCIAL FACTORS ($\Delta R^2$ in %)** | (14.7***) | | | (16.0***) | | | (13.6***) | | |
| **History of mental health problems (Ref = No)** | 0.47 | -0.73, 1.67 | 0.02 | 0.18 | -1.48, 1.84 | 0.01 | 0.40 | -1.34, 2.15 | 0.02 |
| **Sexual orientation (Ref = Heterosexual)** | 0.45 | -0.20, 1.10 | 0.04 | | | | 0.33 | -0.54, 1.19 | 0.03 |
| **Intimate partner violence** | 0.33* | 0.07, 0.59 | 0.08 | 0.38† | -0.05, 0.80 | 0.07 | 0.37* | 0.06, 0.68 | 0.10 |
| **Perceived social support** | -0.10*** | -0.11, -0.08 | -0.38 | -0.12*** | -0.14, -0.09 | -0.43 | -0.11*** | -0.13, -0.09 | -0.44 |
| **COVID-RELATED FACTORS ($\Delta R^2$ in %)** | (5.0***) | | | (6.1***) | | | (11.8***) | | |
| **Tested positive for COVID-19 (Ref = No)** | 0.39 | -1.16, 2.01 | 0.02 | 0.27 | -1.76, 2.30 | 0.01 | | | |
| **Presence of COVID-19 symptoms (Ref = No)** | 0.80† | -0.08, 1.95 | 0.06 | 1.33* | 0.08, 2.59 | 0.09 | 0.62 | -0.54, 1.77 | 0.04 |
| **Other medical conditions (Ref = No)** | 1.12 | -0.26, 2.50 | 0.05 | 1.12 | -0.71, 2.95 | 0.05 | | | |
| **Self-isolation for COVID -19 symptoms (Ref = No)** | 0.79† | -0.06, 1.64 | 0.06 | 0.85 | -0.37, 2.07 | 0.06 | 0.52 | -0.64, 1.67 | 0.04 |
| **Worry about COVID-19 infection** | 0.63*** | 0.35, 0.92 | 0.19 | | | | 0.69** | 0.28, 1.10 | 0.20 |
| **Worry COVID-19 infection in family members** | 0.10 | -0.18, 0.37 | 0.03 | | | | 0.07 | -0.31,0.44 | 0.02 |
| **COVID-19 pandemic perceived as threat to income** | -0.20† | -0.40, 0.01 | -0.06 | -0.19 | -0.49, 0.10 | -0.06 | | | |
| **Feeling isolated during lockdown** | | | | -0.43** | -0.75, -0.12 | -0.12 | 0.50** | 0.21, 0.78 | 0.15 |
| **Disruption from COVID 19 pandemic** | | | | 0.06*** | 0.03, 0.09 | 0.18 | 0.04** | 0.01, 0.07 | 0.14 |

†*p* <0.1;

\**p* <0.05;

\*\**p* <0.01;

\*\*\**p* <0.001;

Beta—standardized regression coefficient.

**Depressive symptoms.** Sociodemographic variables explained 4.4% of the variance in depressive symptoms (Table 5) which was comparable in men and women (4.5% and 5.3% respectively). Male sex and higher educational qualifications were independently associated with lower depressive symptoms.

As with anxiety symptoms, the largest proportion of variance in depressive symptoms was explained by other psychosocial factors (15.6% in the whole sample, and 17.0% and 14.6% in men and women respectively). In the whole sample and among men and women, perceived social support was independently associated with fewer depressive symptoms while intimate partner violence was associated with more depressive symptoms. In contrast

**Table 5. Multivariate associations between depressive symptoms and predictors with sex differences.**

| Predictor variables | Total Coef | 95% CI | Beta | Male Coef | 95% CI | Beta | Female Coef | 95% CI | Beta |
|---|---|---|---|---|---|---|---|---|---|
| **SOCIO-DEMOGRAPHIC FACTORS ($\Delta R^2$ in %)** | (4.4***) | | | (4.5*) | | | (5.3**) | | |
| **Sex (Ref = Female)** | | | | | | | | | |
| Male | -0.59** | -1.02, -0.16 | -0.08 | | | | | | |
| **Age** | -0.02 | -0.05, 0.01 | -0.06 | -0.03 | -0.07, 0.02 | -0.08 | -0.04† | -0.09, 0.00 | -0.11 |
| **Education (Ref = No qualifications)** | | | | | | | | | |
| Primary School | -0.52 | -2.51, 1.48 | -0.02 | -1.60 | -4.00, 0.80 | -0.08 | 1.31 | -2.47, 5.10 | 0.05 |
| Vocational and equivalents | -0.80 | -2.48, 0.88 | -0.05 | -1.06 | -3.04, 0.93 | -0.08 | -0.39 | -3.85, 3.02 | -0.02 |
| Secondary School | -1.71* | -3.31, -0.12 | -0.15 | -1.21 | -3.14, 0.73 | -0.10 | -1.31 | -3.61, 1.83 | -0.13 |
| University | -1.62* | -3.10, -0.14 | -0.19 | -2.03* | -3.77, -0.29 | -0.25 | -0.53 | -4.43, 2.49 | -0.06 |
| **Marital status (Ref = Single)** | | | | | | | | | |
| Married | -0.08 | -0.69, 0.54 | -0.01 | -0.08 | -0.94, 0.82 | -0.01 | 0.24 | -0.63, 1.10 | 0.03 |
| Divorced, Separated or Widowed | 0.79 | -0.39, 1.98 | -0.06 | -0.04 | -0.96, 2.99 | 0.05 | 0.91 | -0.57, 2.40 | 0.05 |
| **Employment status (Ref = Unemployed)** | | | | | | | | | |
| Student | -0.01 | -0.79, 0.68 | 0.01 | -0.06 | -1.15, 0.98 | -0.01 | -0.18 | -1.18, 0.82 | -0.02 |
| Employed | -0.45 | -1.09, 0.20 | -0.06 | -0.02 | -0.94, 0.89 | -0.01 | -0.78† | -1.66, 0.10 | -0.11 |
| **PSYCHOSOCIAL FACTORS ($\Delta R^2$ in %)** | (15.5***) | | | (17.0***) | | | | | |
| **History of mental health problems (Ref = No)** | 1.08* | 0.14, 2.03 | 0.07 | 0.59 | -0.70, 1.88 | 0.04 | 1.46* | 0.02, 2.90 | 0.08 |
| **Sexual orientation (Ref = Heterosexual)** | 0.39 | -0.11, 0.90 | 0.05 | | | | 0.47 | -0.23, 1.18 | 0.05 |
| **Intimate partner violence** | 0.26* | 0.06, 0.46 | 0.08 | 0.45** | 0.12, 0.78 | 0.12 | 0.25† | 0.00, 0.51 | 0.08 |
| **Perceived social support** | -0.08*** | -0.10, -0.07 | -0.41 | -0.08*** | -0.10, -0.06 | -0.38 | -0.09*** | -0.11, -0.07 | -0.45 |
| **COVID-RELATED FACTORS ($\Delta R^2$ in %)** | (5.0***) | | | (0.8) | | | (11.8***) | | |
| **Other medical conditions (Ref = No)** | | | | 1.41* | 0.04, 2.77 | 0.09 | | | |
| **Self-Isolation for COVID -19 Symptoms (Ref = No)** | 0.57† | -0.08, 1.22 | 0.05 | | | | 0.17 | -0.74, 1.08 | 0.02 |
| **Having friend who had COVID-19 (Ref = No)** | -0.61* | -1.20, -0.01 | -0.06 | -0.60 | -1.46, 0.26 | -0.06 | | | |
| **Worry about COVID-19 infection** | 0.38*** | 0.22, 0.54 | 0.14 | | | | 0.20 | -0.04, 0.44 | 0.07 |
| **COVID-19 pandemic perceived as threat to income** | -0.11 | -0.28, 0.05 | -0.04 | 0.05 | -0.17, 0.27 | 0.02 | | | |
| **Feeling isolated during lockdown** | 0.44*** | 0.27, 0.57 | 0.16 | | | | 0.67*** | 0.44, 0.91 | 0.24 |
| **Disruption from COVID 19 pandemic** | | | | | | | 0.05*** | 0.02, 0.07 | 0.19 |

†$p < 0.1$;

*$p < 0.05$;

**$p < 0.01$;

***$p < 0.001$;

Beta—standardized regression coefficient.

to the finding with anxiety symptoms, previous history of psychiatric conditions remained an independent predictor of depressive symptoms with this effect being larger in women compared to men.

In the whole sample, COVID-19-related factors explained 4.5% of the variance in depressive symptoms of which worry about infection and feeling isolated during the lockdown were independently associated with increased depressive symptoms. In men, the proportion of variance explained was 0.8% with the presence of COVID-19-risk conditions being independently associated with depressive symptoms. For women, these group of symptoms accounted for a larger 11.8% of the variance in depressive symptoms with worry about infection, isolation during lockdown and disruption due to the COVID pandemic emerging as independent predictors of depressive symptoms among them.

## Discussion

The present study aimed to investigate factors associated with anxiety and depressive symptoms during the COVID-19 pandemic, and to test for sex differences in these relationships. This is the first study to investigate these relationships among Nigerians. We demonstrated significant associations between sociodemographic, psychosocial and COVID-19 related factors and anxiety and depressive symptoms. The variances in anxiety and depressive symptoms explained by COVID-19-related factors were larger in women compared to men.

In contrast to the high global rates of COVID-19 infection [36, 37], only five percent of the sample reported testing positive while higher proportions reported either having had COVID-19 symptoms or self-isolating on account of these or knowing people who had tested positive or had died from COVID-19. These contrasting rates are consistent with the low rates of testing that have been described for COVID-19 in Nigeria [21, 38, 39] and suggest either a low uptake of testing by Nigerians or a limited capacity of testing services. The psychosocial burden of the pandemic was indicated by the large proportions of participants who indicated being worried about themselves or family members developing COVID-19, feeling isolated during the temporary lockdown instituted in Nigeria [40, 41] and who perceived the pandemic as a threat to their income [42]. While the rates of testing and presence of COVID-19 symptoms were comparable in male and female participants, women reported more worry about personal and family risk for COVID-19 infection and the impact of the lockdown on their incomes. This is consistent with previous findings [43] and may reflect the higher pre-pandemic risk for stress-related conditions in women [25, 44]. In contrast, men reported greater disruption in their lives due to the pandemic and this may be understood in the context of the male participants in the present study being older and more likely to be married. They may, thus, be more sensitive to disruptions and consequent threats to their roles as breadwinners.

Consistent with findings from previous research [24, 25, 44], depressive and anxiety symptoms were significantly higher among female participants. These may reflect the impacts of higher levels of psychosocial stressors among women such as intimate partner violence or higher COVID-19-related concerns among them such as worry about infection [43].

### Associations with anxiety and depressive symptoms

Of the sociodemographic variables, being employed, higher income and higher education were associated with lower anxiety and depressive symptoms. suggesting that higher socioeconomic status may provide a buffer against economic uncertainties impacts during the COVID-19 pandemic [45–48].

Consistent with prior findings, previous mental illness [49], non-heterosexuality [10, 11] and intimate partner violence [8, 13] were individually associated with higher anxiety and depressive symptoms while perceived social support was associated with lower anxiety and depressive symptoms [18, 50]. Together, these variables explained the largest proportion of variance in both anxiety and depressive symptoms with intimate partner violence and perceived social support retaining independent associations with both symptoms. These suggest that the previously reported mental health disparities in Nigerian non-heterosexual compared with heterosexual men [11] may be partly explained by psychosocial disadvantages [51], although this needs to be specifically investigated. Prior mental illness was only independently associated with depressive symptoms and may reflect other mechanisms such as disruption in mental health services [21] or greater neurological vulnerability to stressors [49].

As have been previously demonstrated, most of the COVID-19-related factors including testing positive for COVID-19 infection, having COVID-19 symptoms, having to self-isolate, worry about getting infected were significantly associated with higher anxiety symptoms [1, 2].

In contrast, fewer factors were individually associated with depressive symptoms—increased isolation during lockdown and in the broader context of the pandemic, and concern about getting infected. However, the variance explained in both psychopathologies by COVID-19-related factors was comparable. The independent association between worry about infection and both anxiety and depressive symptoms after adjusting for sociodemographic and psychosocial factors indicates its significance as a stressor. Isolation during lockdown was further independently associated with increased depressive symptoms which is consistent with previous research [18]. The attenuation of the associations between the other COVID-19 factors and anxiety and depressive symptoms after adjusting for other socio-demographic and psychosocial risk and protective factors suggest that the mental health impacts of these pandemic-related factors may be partly explained by indirect effects through these processes. Thus, while neurotoxic effects of the virus have been suggested as a possible mechanism for psychopathology in the context of the COVID-19 pandemic [3, 4]; our findings suggest that alternative pathways including exacerbation of pre-existing psychosocial risks such as intimate partner violence victimization [52], increased minority stress among non-heterosexual Nigerians [21] and attenuation of protective factors such as social support may explain the associations between pandemic-related stressors and mental health problems; however, these need to be specifically tested.

## Sex differences

An important sex difference was that COVID-19-related factors explained a larger proportion of the variance in anxiety and depressive symptoms in female compared to male participants. This was most marked for depressive symptoms in which the variance explained in male participants was very small and not statistically significant. This suggests that the psychosocial risk and protective factors investigated in the present study respectively completely explain or mitigate the risk conferred by the COVID-19 pandemic in Nigerian men. In contrast, other mechanisms which were not investigated in the present study may further explain the associations between anxiety and depressive symptoms among women in Nigeria. These may include biological [25, 53] or social vulnerabilities [26] especially in low/middle income countries like Nigeria with high levels of gender inequality [27]. Alternatively, it is also possible that men react to stress with externalising behaviors such as substance misuse [24] which were not assessed in the present study.

## Conclusion

Our findings indicate that some psychosocial factors related to the COVID-19 pandemic including worry about infection, disruption due to the pandemic, and isolation during the lockdown are independently associated with higher depressive and anxiety symptoms. In contrast, higher education, being employed, higher income and higher perceived social support were associated with lower anxiety and depressive symptoms suggesting that higher socioeconomic status and social support may be protective against the uncertainties and adverse mental health impacts of the COVID-19 pandemic in Nigeria. Thus, improving social support can be potentially targeted as an intervention to minimize the adverse mental health impacts of the COVID-19 pandemic in resource-poor setting such as Nigeria, however, this needs to be tested using prospective and experimental designs. Such support may be better targeted at individuals from lower socioeconomic backgrounds. Similarly, women in Nigeria may benefit more from such interventions; however, more research is needed to identify other mechanisms of increased depressive and anxiety symptoms among them in the context of the COVID-19 pandemic.

## Limitations

In interpreting our findings, the following limitations should be considered: the online nature of survey may have limited representativeness of the sample, for example, individuals with higher levels of education were over-represented, as were Yorubas who reside in southwestern Nigeria. Although this may limit the generalizability of our findings, the better socioeconomic indices including higher education and average income in South-Western Nigeria [54] may mean that the associations between COVID-19-related factors and anxiety and depressive are underestimated in our study. Future studies may overcome this by using translated question-naires and specifically recruiting participants with lower levels of formal education. The cross-sectional nature of the study also means that our findings cannot be extended to explain causal mechanisms which would need to be investigated using prospective or experimental designs.

## Acknowledgments

We acknowledge Prof. Morenike. Folayan for her suggestions to improve the manuscript and we thank the study participants.

## Author Contributions

**Conceptualization:** Olakunle Ayokunmi Oginni, Ibidunni Olapeju Oloniniyi, Olanrewaju Ibigbami, Victor Ugo, Adedotun Ogunbajo, Boladale Mapayi.

**Formal analysis:** Olakunle Ayokunmi Oginni, Ayomipo Amiola, Adedotun Ogunbajo, Ola-doyin Esan.

**Resources:** Ibidunni Olapeju Oloniniyi, Olanrewaju Ibigbami, Victor Ugo, Aderopo Adelola, Oluwatosin Daropale, Matthew Ebuka.

**Supervision:** Boladale Mapayi.

**Writing – original draft:** Olakunle Ayokunmi Oginni.

**Writing – review & editing:** Olakunle Ayokunmi Oginni, Ibidunni Olapeju Oloniniyi, Olan-rewaju Ibigbami, Victor Ugo, Ayomipo Amiola, Adedotun Ogunbajo, Oladoyin Esan, Aderopo Adelola, Oluwatosin Daropale, Matthew Ebuka, Boladale Mapayi.

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
