## [Decision Letter · Decision Letter 0]

21 Jul 2021

PONE-D-20-39501

Depressive and anxiety symptoms and COVID-19-related factors among men 1and women in Nigeria

PLOS ONE

Dear Dr. Oginni,

Thank you for submitting your manuscript to PLOS ONE. After careful consideration, we feel that it has merit but does not fully meet PLOS ONE’s publication criteria as it currently stands. Therefore, we invite you to submit a revised version of the manuscript that addresses the points raised during the review process.

We look forward to receiving your revised manuscript.

Kind regards,

Associate Professor Dr Muhammad Aziz Rahman

Academic Editor

PLOS ONE

Journal Requirements:

Reviewers' comments:

Reviewer's Responses to Questions

**Comments to the Author**

1. Is the manuscript technically sound, and do the data support the conclusions?

Reviewer #1: Yes

Reviewer #2: Yes

2. Has the statistical analysis been performed appropriately and rigorously? 

Reviewer #1: Yes

Reviewer #2: Yes

3. Have the authors made all data underlying the findings in their manuscript fully available?

Reviewer #1: Yes

Reviewer #2: Yes

4. Is the manuscript presented in an intelligible fashion and written in standard English?

Reviewer #1: Yes

Reviewer #2: Yes

5. Review Comments to the Author

Reviewer #1: Review Comments to the Author

Please use the space provided to explain your answers to the questions above. You may also include additional comments for the author, including concerns about dual publication, research ethics, or publication ethics

My comments: Line 121

- More information needed on the tool – is it a validated tool?

- Any piloting?

- More details needed

- The sample size is not representative or limited – Rationalise explain the reason

Female male distribution needs more input: material and method section, data analysis and also in the results section

Reviewer #2: The study addresses an area of research not previously investigated in Nigeria, nor in the context protective factors for mental well-being during COVID-19.

The paper is well written and likely to be of interest to a wide audience.

The demographics of the participants were a little surprising, the anxiety and depressive symptoms were more as anticipated.

P18, line 316 'To the authors knowledge' could be removed as it does not carry much weight and earlier in the paper,( P3) the authors have stated that this is the first paper on the topic in Nigeria

The recommendation for more research specifically on women with increased anxiety and depression symptoms is hopefully pursued.

6. PLOS authors have the option to publish the peer review history of their article (what does this mean?). If published, this will include your full peer review and any attached files.

Reviewer #1: No

Reviewer #2: **Yes: **Virginia Plummer

---

## [Author Response · Author response to Decision Letter 0]

27 Jul 2021

Dear Prof. Rahman,

Re: Depressive and anxiety symptoms and COVID-19-related factors among men and women in Nigeria. Manuscript number: PONE-D-20-39501

Thank you for your kind consideration of our above-titled manuscript for publication in your esteemed journal. We are also grateful for the feedback from yourself and the reviewers which have made us think further about our findings and improve the quality of our manuscript.

To make our manuscript compatible with PLOS ONE’s style requirements, we have made the following changes:

a. The symbols in the author list have now been changed to those recommended by PLOS ONE.

b. The wording of the author contribution has been made consistent with the format recommended by PLOS ONE.

c. The font sizes of level 2 and level 3 headings have now been made 16 and 14 respectively.

d. The dois in the reference list have also been updated to the latest format.

e. The Acknowledgment section has been moved to the end of the manuscript – between the discussion and reference list.

f. The font of the text in the tables has been changed to Times New Roman and the font size increased to 10 to improve legibility.

We have also gone through the reference list and none of the cited articles have been retracted. A preprint has been accepted for publication and this has been updated (Yan et al., 2021). Four references have also been included (Reuben et al., 2021; Habib et al., 2021; Ogunbajo et al., 2020). These have been appropriately included as in-text citations and in the reference list.

Please find below our responses to the reviewers’ comments. For ease of presentation, the reviewers’ comments are presented in bold type, while our responses are presented in regular type and excerpts from the main manuscript are presented in italics.

We hope that the changes made and our responses to the reviewers’ comments are satisfactory.

Best wishes,

Olakunle Oginni

(Corresponding author)

 

Reviewer #1

My comments: 

1. Line 121: More information needed on the tool – is it a validated tool? Any piloting? More details needed

Thank you for this comment. These questions have been previously used to assess these sociodemographic variables among Nigerians [1, 2]. We therefore felt that there was no need to pilot the questions as they had been previously used in the study population. We have now amended this section to indicate that these questions have been previously used among Nigerians as follows: “These questions have been previously used to assess sociodemographic information among Nigerians [11, 31] and were therefore considered appropriate.”, lines 130-132.

2. The sample size is not representative or limited – Rationalise explain the reason

Thank you for highlighting this concern. The sample size was considered adequate based on online surveys carried out among Nigerians during the COVID-19 pandemic with sample sizes ranging from 589 [3] to 886 [4]. We have now included the following sentence in the manuscript to justify our sample size: “This gave a total 966 participants which was larger and comparable to other online surveys which were carried out in Nigeria during the COVID-19 pandemic [29, 30].”, lines 117-119. 

Regarding the representativeness of the sample, we think the high proportion of participants with higher education may be related to the online nature of the survey. The inclusion criteria (such as being fluent in English and being able to use the internet; lines 105-106), which were partly meant to reduce non-valid responses and increase the validity of the study, may have further contributed to this. We have recognised this effect in the limitations section as follows: “…the online nature of survey may have limited representativeness of the sample, for example, individuals with higher levels of education were over-represented…”, lines 431-433. We also note that this may “…limit the generalizability of our findings…”, line 434.

Considering that the sex ratio of the sample is comparable to that in the population and that there is sufficient diversity with respect to the other sociodemographic variables (e.g., socioeconomic status, age, marital status and employment status), we believe that the present study can still contribute to understanding the associations between COVID-19-related stressors and anxiety and depressive symptoms in low-and middle-income settings. We have now stated that “Future studies may overcome this by using translated questionnaires and specifically recruiting participants with lower levels of formal education.”, lines 437-439.

3. Female male distribution needs more input: material and method section, data analysis and also in the results section

We are grateful for this suggestion. We have made the following changes:

In the material and method section, we have now indicated that ‘…male and female participants had equal chances of participating in the survey.”, lines 113-114. 

In the analyses section, we have also indicated how we determined sex differences in the univariate and multivariable regression analyses as follows: “For the univariate and multivariate regression analyses, sex differences in the regression coefficients were determined by inspecting the 95% confidence intervals of the coefficients in male and female participants and differences were deemed to be statistically significant when the confidence intervals did not overlap. Similarly, sex differences in ΔR2 in multivariate regression models were determined by comparing the magnitudes of ΔR2 in males and female participants.”, lines 196-201.

In the results section, we had previously incorporated the description of sex differences into each section of the results to maintain the flow; we have now indicated in the subtitles that each section includes sex differences.

We have also included a subheading in the discussion to indicate sex differences.

Reviewer #2

The study addresses an area of research not previously investigated in Nigeria, nor in the context protective factors for mental well-being during COVID-19. The paper is well written and likely to be of interest to a wide audience.

We are grateful for this feedback.

The demographics of the participants were a little surprising, the anxiety and depressive symptoms were more as anticipated.

We agree with this and suggest that it may reflect bias related to the online nature of the survey. This has been specifically highlighted in the Limitations section.

P18, line 316 'To the authors knowledge' could be removed as it does not carry much weight and earlier in the paper,( P3) the authors have stated that this is the first paper on the topic in Nigeria

Thank you for this suggestion. This has been removed.

The recommendation for more research specifically on women with increased anxiety and depression symptoms is hopefully pursued.

This is noted with thanks.

References

1. Oginni OA, Mosaku KS, Mapayi BM, Akinsulore A, Afolabi TO. Depression and associated factors among gay and heterosexual male university students in Nigeria. Arch Sex Behav. 2018;47(4):1119-1132. https://doi.org/10.1007/s10508-017-0987-4, PMID: 28466230

2. Ogunbajo A, Oginni OA, Iwuagwu S, Williams R, Biello K, Mimiaga MJ. Experiencing intimate partner violence (IPV) is associated with psychosocial health problems among gay, bisexual, and other men who have sex with men (GBMSM) in Nigeria, Africa. J Interpers Violence. 2020; https://doi.org/10.1177/0886260520966677, PMID: 33118468

3. Reuben RC, Danladi MM, Saleh DA, Ejembi PE. Knowledge, attitudes and practices towards COVID-19: An epidemiological survey in North-Central Nigeria. J Community Health. 2021;46(3):457-470, https://doi.org/10.1007/s10900-020-00881-1, PMID: 32638198

4. Habib MA, Dayyab FM, Iliyasu G, Habib AG. Knowledge, attitude and practice survey of COVID-19 pandemic in Northern Nigeria. PLOS ONE. 2021;16(1):e0245176, https://doi.org/10.1371/journal.pone.0245176, PMID: 33444360

---

## [Editor Report · Decision Letter 1]

13 Aug 2021

Depressive and anxiety symptoms and COVID-19-related factors among men and women in Nigeria

PONE-D-20-39501R1

Dear Dr. Oginni,

We’re pleased to inform you that your manuscript has been judged scientifically suitable for publication and will be formally accepted for publication once it meets all outstanding technical requirements.

Kind regards,

Associate Professor Dr Muhammad Aziz Rahman

Academic Editor

PLOS ONE

---

## [Editor Report · Acceptance letter]

18 Aug 2021

PONE-D-20-39501R1 

Depressive and anxiety symptoms and COVID-19-related factors among men and women in Nigeria 

Dear Dr. Oginni:

I'm pleased to inform you that your manuscript has been deemed suitable for publication in PLOS ONE. Congratulations! Your manuscript is now with our production department. 

Kind regards, 

on behalf of

Associate Professor Dr. Muhammad Aziz Rahman 

Academic Editor

PLOS ONE